# Research on Geometric Parameters Optimization of Fixed Frog Based on Particle Swarm Optimization Algorithm

**Rang Zhang** * , **Gang Shen and Xujiang Wang**

Institute of Rail Transit, Tongji University, Shanghai 201804, China
* Correspondence: zhangrang@tongji.edu.cn

**Abstract:** In this paper, to improve the wheel/rail dynamic performance of the vehicle passing through the fixed frog area and improve the service life of the fixed frog, a geometric parameter optimization design method of the fixed frog area is proposed using particle swarm optimization (PSO). Based on the variable section rail profile interpolation algorithm and wheel/rail contact solution algorithm, the wheel/rail contact characteristics of the fixed frog area are analyzed. Then, the vehicle-fixed frog dynamic model is built using a MATLAB/Simulink platform to complete the dynamic calculation and analysis of the rail vehicle passing through the fixed frog area. Finally, based on the wheel/rail contact characteristics of the fixed frog area, and take wheel/rail forces as the optimization goal, the optimization design method for the wing rail lifting value and the nose rail height of the fixed frog area is proposed. The comparative analysis shows that the wheel/rail dynamic performance in the fixed frog area has been greatly improved after optimization, which verifies the feasibility of the optimization strategy.

**Keywords:** fixed frog; wheel/rail interaction; geometric parameter optimization; PSO

## 1. Introduction

Turnout is the key part of the track structure of the railway system, and its role is to guide the rail vehicle from one track to another, so as to give full play to the transportation efficiency of the line. However, due to its complex structure, large number of applications, short service life, low traffic safety, limited train speed and high maintenance cost, it brings a great workload to the daily maintenance and overhaul of the railway public works department. Therefore, turnout, curve and joint are called the three weak links of track [1]. As the fixed frog turnout is cheaper and more reliable than the movable core rail turnout, it is widely used in a large number of the common railway, the heavy-haul line and the urban rail transit line [2].

The fixed frog turnout mainly includes switch parts, connecting parts and frog area parts. Among them, the gap from the throat of the frog to the actual point of nose rail is called the harmful space of a fixed frog. The wheel/rail relationship of the wheelset passing through a harmful space is very complex, which is also the main reason for limiting the vehicle crossing speed [3].

Due to the harmful space, the wheel/rail force in the frog is much greater than that in the general section line. This violent dynamic interaction becomes more and more serious with the speed increase of the train. This can not only cause strong wheel/rail impact and even derailment of vehicles, but also cause serious wear, fracture and other turnout diseases of the nose rail in the frog section. Therefore, it is particularly essential to research the wheel/rail interaction relationship of vehicles passing through the fixed frog area.

Many scholars have carried out extensive research on reducing wheel/rail impact in a fixed frog area. Ren [4,5] established the vehicle-frog vertical vibration model, calculated the wheel/rail impact force in the fixed frog through a self-made program, carrying out the wheel/rail dynamics research first in the fixed frog area in China. Wang [6,7] regarded

the fixed frog as a finite length variable cross-section Euler beam, established a vehicle-turnout spatial coupling vibration model including the vibration of the fixed frog, the connecting part and the switch, and comprehensively research the influence of vehicle and track parameters on the wheel/rail relationship in the turnout area. Lagos [8] used four different treads and two different turnouts to conduct a dynamic simulation analysis of vehicle crossing, and results show the geometric design for turnouts play a very important role on the dynamic performance of vehicle crossing. Markine [9,10] established a complete dynamic model of the fixed frog to research the fatigue failure of the nose rail under the high-frequency wheel/rail impact load, analyzed the influence of the stiffness of the lower support structure in the fixed frog area on the contact fatigue failure of the nose rail, and found that the use of the under-rail pad with small stiffness is helpful to decrease the impact force on the rail. Grossoni [11] iteratively optimized the stiffness under the track in the fixed frog area based on Genetic Algorithm to reduce the common failure forms such as fatigue failure of the fixed frog. Anderson [12,13] established the wheel/turnout contact finite element model considering the characteristics of track variable stiffness, rail variable section and wheel/rail multi-point contact in detail, and analyzed the dynamic performance of the wheel/turnout in a wide frequency range. Blanco-Saura [14] built a detailed 3D finite element model for turnout by using ANSYS software, and combined with the vehicle dynamics model established by VAMPIRE Pro, analyzed the vertical dynamic response of turnout under dynamic load, especially the vertical dynamic force characteristics of center rail and switch rail area.

In the research on the optimization method for the wing rail lifting value and the nose rail height, Palsson [15] took the wheel/rail contact stress as the optimal goal, and proposes a cross geometric design strategy for the crossing to optimize the rail shape in the frog area, so as to reduce the rail damage. Wan [16] presented an optimization strategy for deceasing wheel/rail wear through changing nose rail profiles. Cao [17,18] obtained a more reasonable reduction value of a key section of nose rail through researching the wheel/rail static parameters and wheel/rail dynamic action performance in a fixed frog area. Xu [19] obtained a fixed frog optimization design method based on the ratio of the required height difference of wheel profile to the actual height difference of rail top profile in the fixed frog. Zhang [20] focused on the effects of the wing rail height value in fixed frog area on driving stability and wheel/rail dynamic action when vehicles pass through the turnout center.

Although the existing research has done a lot of in-depth and continuous work, there are still shortcomings. When the number of control sections of fixed frogs is large, the arrangement and combination schemes increase rapidly, which leads to the extremely low efficiency of trial and error method and is basically not feasible. Therefore, it is a difficult problem to propose a complete and standardized design process for the optimization design of key geometric parameters of the fixed frog area. This paper will focus on the wheel/rail interaction performance of rail vehicles passing through the fixed frog area, and decrease the wheel/rail impact by optimizing and improving geometric structure in the frog area, especially the wing rail lifting value and the nose rail height, so as to provide some theoretical basis for turnout design, application and maintenance. Additionally, to change the traditional design method of repeated trial and error and then dynamic verification, based on the comprehensive analysis of the wheel/rail contact relationship in the fixed frog area, a closed-loop design method of the geometric parameter optimization in the fixed frog area is proposed by using the particle swarm optimization algorithm, which has great engineering application value.

## 2. Wheel/Rail Contact Analysis in the Fixed Frog Area

To comprehensively explore the wheel/rail contact characteristics in a fixed frog area, variation laws of key wheel/rail contact parameters under different wheelset lateral displacement are analyzed; see Figure 1 for the calculation results. Figure 1a,b show that the wheelset passing the fixed frog area, the lateral coordinate of the wheel/rail contact

points gradually increases from the throat with the wing rail continuously deviating from the gauge center line, and the vertical coordinate gradually decreases with the wing rail continuously height. This means that during the contact between the wing rail and wheel, wheel/rail contact points transfer from near the wheel profile rolling circle to the side away from the flange. When the wheelset is about 240 mm away from theoretical point of frog, the wheelset load begins to transfer to the nose rail, and coordinates of the wheel/rail contact point suddenly change. Additionally, wheel/rail contact point lateral coordinates instantaneously decrease and the vertical coordinates instantaneously increase. Then, with the continuous widening and lifting of the nose rail, lateral coordinates of wheel/rail contact points continue to increase, and the vertical coordinates continue to decrease and gradually return to the initial value.

From the perspective of the wheelset lateral shift, with the wheelset lateral shift increasing and moving towards the nose rail, the smaller the lateral coordinate jump amplitude of wheel/rail contact coordinate points and the larger vertical coordinate jump amplitude. At the same time, it also means that the shorter the contact time between the wing rail and wheel, the position of the section where the nose rail begins to bear load is closer to the theoretical point. Figure 1c shows that the rolling circle radius increases to more than 440 mm instantaneously, which means that the wheel flange contact occurred, resulting in a sharp increase in the vertical contact coordinate points and a risk that the wheelset will climb to the nose rail surface.

The above analysis shows that when the wheelset load is moved from the wing rail to the nose rail, the wheel will jump. If the wing rail height value is set unreasonably, it will directly lead to a large jump of the wheel, thus deteriorating the wheel/rail contact performance in the frog. In addition, to strictly guarantee the safety of the vehicle and prevent the wheelset from climbing the track in the wheel load transfer stage, the geometric structure design of the fixed frog should ensure that the wheelset passes through the turn-out center with a small lateral displacement.

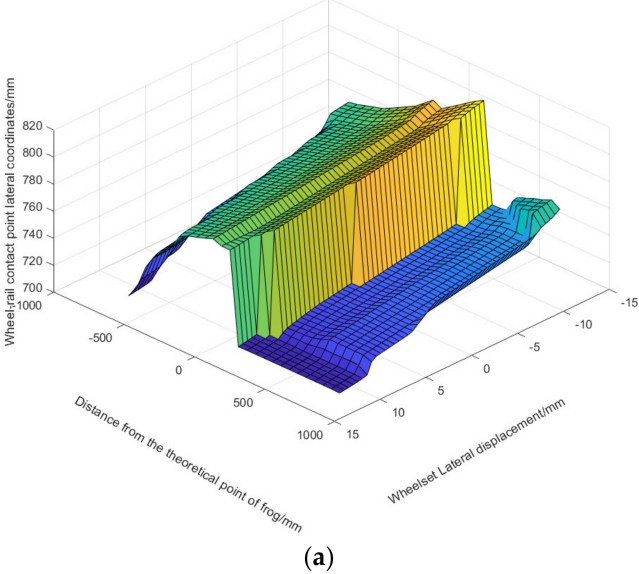

(**a**)

**Figure 1.** *Cont.*

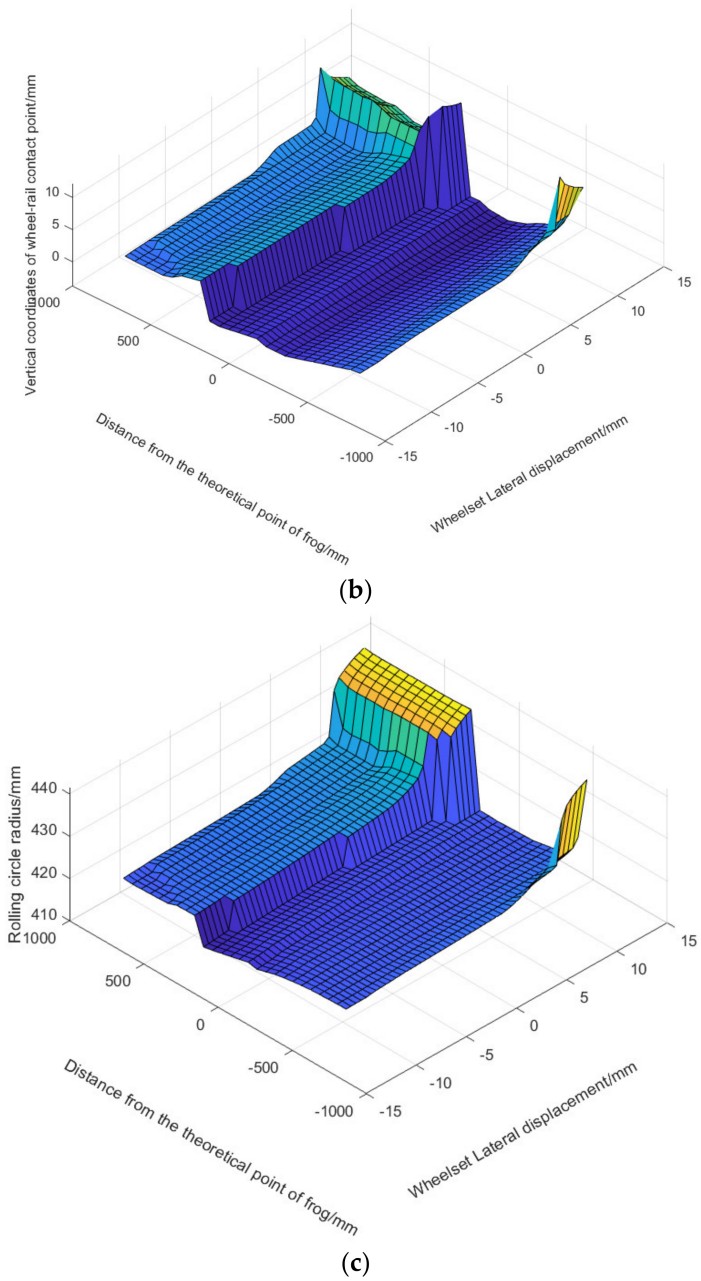

(b)

(c)

**Figure 1.** Wheel/rail contact calculation results of key sections: (**a**) Lateral coordinates of wheel/rail contact point; (**b**) Vertical coordinates of wheel/rail contact point; (**c**) Rolling circle radius.

## 3. Dynamics Modeling of Vehicle/Fixed Frog Interaction

### 3.1. Vehicle Dynamics Modeling

A dynamic model for the four-axle rail vehicle is built to more accurately simulate the dynamic characteristics of the rail vehicle passing the fixed frog area. The frog bodies of the vehicle dynamics system include 1 car body, 2 frames, and 4 wheelsets. In addition, the suspension system includes primary suspension and secondary suspension as shown in Figure 2. Assuming the vehicle runs at a uniform speed, the telescopic vibration of the wheelset, frames, and car body are negligible. Each rigid body considers 5 degrees of freedom (DOF), a total of 35 DOFs as shown in Table 1.

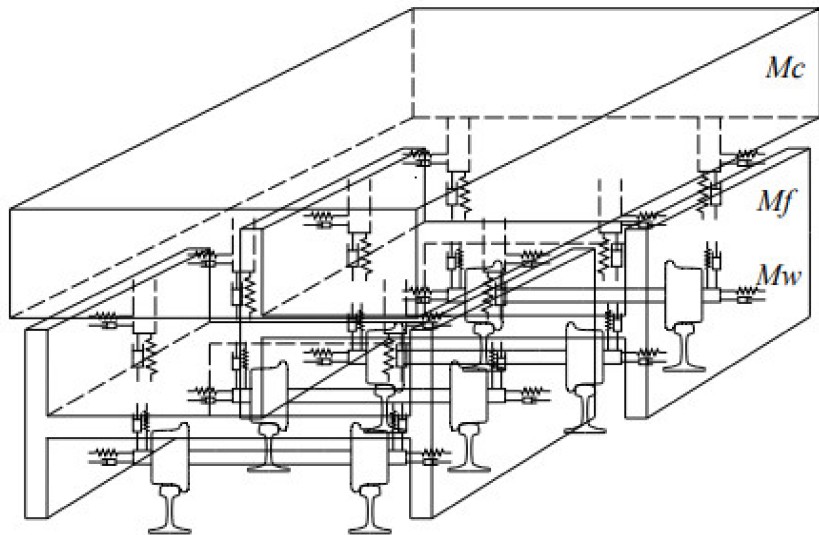

**Figure 2.** Calculation diagram of four-axle rail vehicle.

**Table 1.** Model DOFs.

| Rigid Body | Lateral Motion | Vertical Motion | Rolling Motion | Pitch Motion | Yaw Motion | / |
|------------|----------------|-----------------|----------------|--------------|------------|---|
| Car body | $Y_c$ | $Z_c$ | $\Phi_c$ | $\Theta_c$ | $\Psi_c$ | / |
| Frames | $Y_{bn}$ | $Z_{bn}$ | $\Phi_{bn}$ | $\Theta_{bn}$ | $\Psi_{bn}$ | $n = 1, 2$ |
| Wheelsets | $Y_{wi}$ | $Z_{wi}$ | $\Phi_{wi}$ | $\Theta_{bi}$ | $\Psi_{wi}$ | $i = 1, 2, 3, 4$ |

Based on the matrix assembly method [21], the general representation of the dynamic equation for the railway vehicle is:

$$\mathbf{M}\{\ddot{\mathbf{q}}\} + \mathbf{C}\{\dot{\mathbf{q}}\} + \mathbf{K}\{\mathbf{q}\} = \{\mathbf{F}\}$$

where, $\{\mathbf{q}\}$ is the DOF vector of the model;

$\mathbf{M}$ is the inertia matrix;

$\mathbf{C}$ is the damping matrix;

$\mathbf{K}$ is the stiffness matrix;

$\{\mathbf{F}\}$ is the external force matrix, ignoring the track irregularity, which is mainly composed of 3 parts:

$$\mathbf{F} = \mathbf{F}_{wr} + \mathbf{F}_{IG} + \mathbf{F}_c$$

The meaning of each force component is as follows:

$\mathbf{F}_{wr}$ is the force transmitted by the rail to the wheelset, including normal force and tangential force (creep force);

$\mathbf{F}_{IG}$ is gravity, inertia force, and Coriolis force;

$\mathbf{F}_c$ is the zero position non-equilibrium additional force due to curve line conditions.

This paper uses MATLAB/Simulink to build the railway vehicle dynamic model for the subsequent optimization solution, as shown in Figure 3.

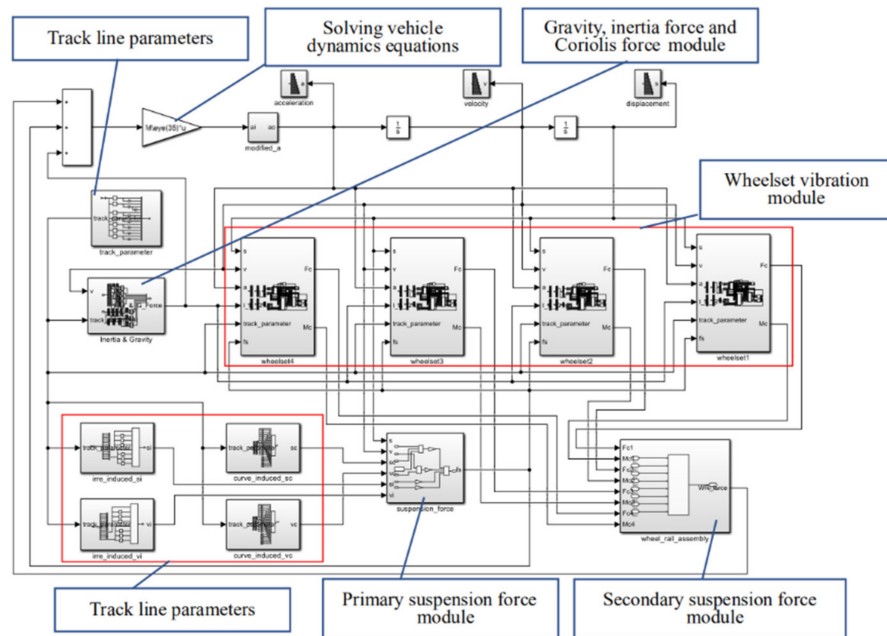

**Figure 3.** Vehicle dynamics model based on Simulink.

### 3.2. Rail Vibration System Model

According to the different problems studied, the rail dynamics models selected are different, mainly including the massless rail, the inertial rail, and the flexible track. Among them, the inertial rail model can be for simulation of complex wheel/rail contact: railway track dynamic performance in the switches and turnouts, contacts between wheel-back and rail, vehicle derailment, etc., so the inertial rail model is selected for dynamic calculation in the paper. Assuming wheel/rail contact of a certain section on one side, the rail is simplified as a mass block with 3 DOFs: lateral shift, vertical shift and torsion, and the mass block is connected with the foundation through equivalent spring and damping. Selecting right rail as an illustration, the dynamic modeling of the rail vibration system is carried out as shown in Figure 4.

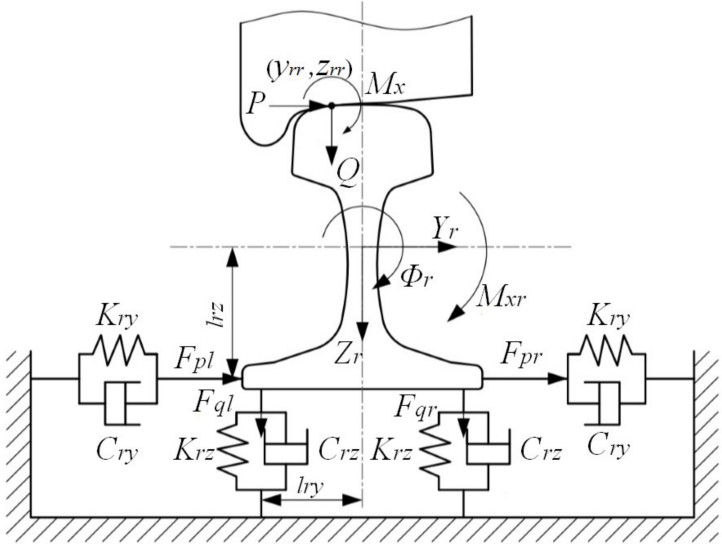

**Figure 4.** Equivalent model of track vibration model.

The coordinate of wheel/rail contact point is $(y_{rr}, z_{rr})$ in the rail coordinate system. P is the lateral component, Q is the vertical component, and $M_x$ is the roll moment, respectively.

$K_{ry}$ and $C_{ry}$ are the lateral support stiffness and damping of sleeper, fastener, and other basic components to the rail. $K_{rz}$ is the vertical foundation stiffness of sleeper, fastener, and other basic components to the rail, and $C_{rz}$ is the damping. Therefore, the fulcrum reaction of the rail is shown in Equation (1).

$$\begin{cases} F_{ql} = -K_{rz}(z_r - \phi_r l_{ry}) - C_{rz}(\dot{z}_r - \dot{\phi}_r l_{ry}) \\ F_{qr} = -K_{rz}(z_r + \phi_r l_{ry}) - C_{rz}(\dot{z}_r + \dot{\phi}_r l_{ry}) \\ F_{pl} = -K_{ry}(y_r - \phi_r l_{rz}) - C_{ry}(\dot{z}_r - \dot{\phi}_r l_{rz}) \\ F_{pr} = -K_{ry}(y_r - \phi_r l_{rz}) - C_{ry}(\dot{z}_r - \dot{\phi}_r l_{rz}) \end{cases} \tag{1}$$

The torsional torque of the rail is shown in Equation (2):

$$M_{xr} = -K_{\phi r}\phi_r - C_{\phi r}\dot{\phi}_r \tag{2}$$

where, $K_{\phi r}$ is the rail torsional stiffness; $C_{\phi r}$ is the rail torsional damping.

Further, the vibration expression for the rail system can be obtained as shown in Equation (3):

$$\begin{cases} m_r \ddot{y}_r = P + F_{pl} + F_{pr} \\ m_r \ddot{z}_r = Q + F_{ql} + F_{qr} + m_r g \\ I_{rx}\ddot{\phi}_r = M_x + Py_{rr} - Qz_{rr} + M_{xr} - F_{pl}l_{rz} - F_{pr}l_{rz} + F_{qr}l_{ry} - F_{ql}l_{ry} \end{cases} \tag{3}$$

where, $m_r$ is the equivalent mass of rail, $y_r$, $z_r$, and $\phi_r$ are lateral, vertical shift, and torsional angle of the rail.

The inner back of the wheelset impacts the side of the guard rail, resulting in additional wheel/rail impact force. Due to the short action time and large amplitude of this wheel/rail impact force, it can be considered as the lateral impact force between wheel-back and guard rail [16].

To simulate the instantaneous impact load between wheel-back and guard rail, a contact condition is equivalent to the spring with great stiffness. Therefore, the wheel/rail elastic compression equation is:

$$\varepsilon = l_{wc} - l_{tc} + y_{rail} + 0.009 + d_a - \Delta_a(x) \tag{4}$$

where, $k_a$ is the equivalent spring stiffness, $l_{wc}$ is the lateral distance between wheel/rail contact point and the origin in wheelset translation coordinate system, $l_{tc}$ is the lateral distance between wheel/rail contact point and the origin in the rail coordinate system, $y_{rail}$ is the dynamic traverse of stock rail, $d_a$ is the wheel flange thickness, $\Delta_a(x)$ is the wheel flange groove width at current position of the wheelset. Further, the wheel/rail impact force is:

$$F_a = \begin{cases} k_a \varepsilon, & \varepsilon > 0 \text{ (Contact state)} \\ 0, & \varepsilon \leq 0 \text{ (Non contact state)} \end{cases} \tag{5}$$

### 3.3. Real-Time Calculation for the Wheel/Rail Interaction

For the optimization problem in the paper, real-time calculation is indispensable. The transmission of wheel/rail creep force and normal forces between vehicle vibration system and rail vibration system are completed through the wheel/rail interaction module.

Figure 5 shows the calculation process of wheel/rail interaction. Firstly, calculate the wheel/rail real-time contact based on the wheelset and rail motion state at time $t$. Then, calculate wheel/rail creep rate, normal forces and contact forces between wheel-back and guard rail based on the wheel/rail real-time contact parameters and the wheel/rail real-time contact parameters, then calculate wheel/rail contact spots and creep coefficient based on the normal force and wheel/rail real-time contact parameters, and then calculate

the creep force. Finally, the above creep forces, normal forces, and lateral impact forces are input into the vehicle model and the rail vibration system respectively, and motion states of the wheel and rail at $t + 1$ time is obtained, which starts again and again until the vehicle runs to the end of the frog. Based on the above analysis, Simulink is used to build a real-time calculation module for the wheel/rail interaction as shown in Figure 6.

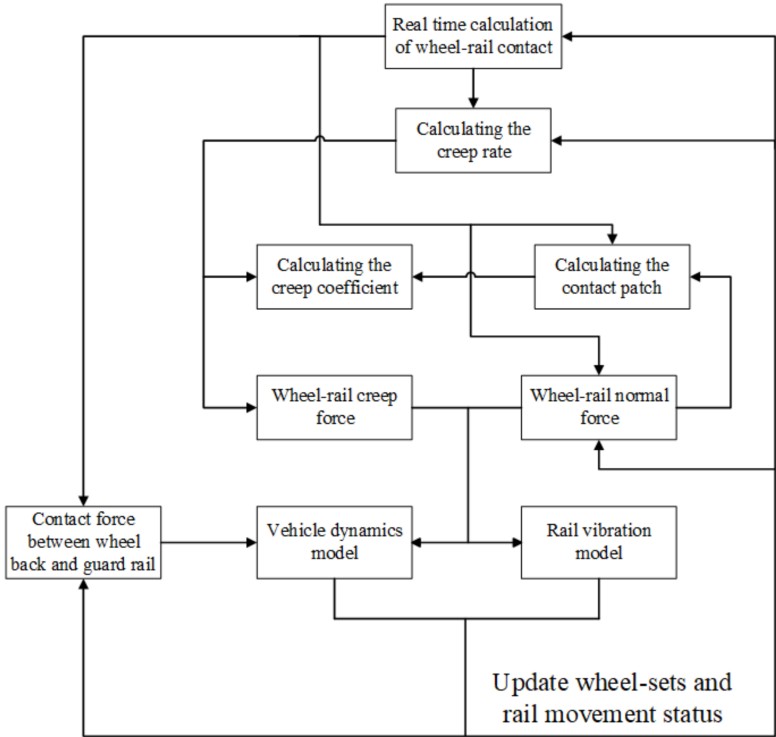

**Figure 5.** Calculation process of wheel/rail interaction.

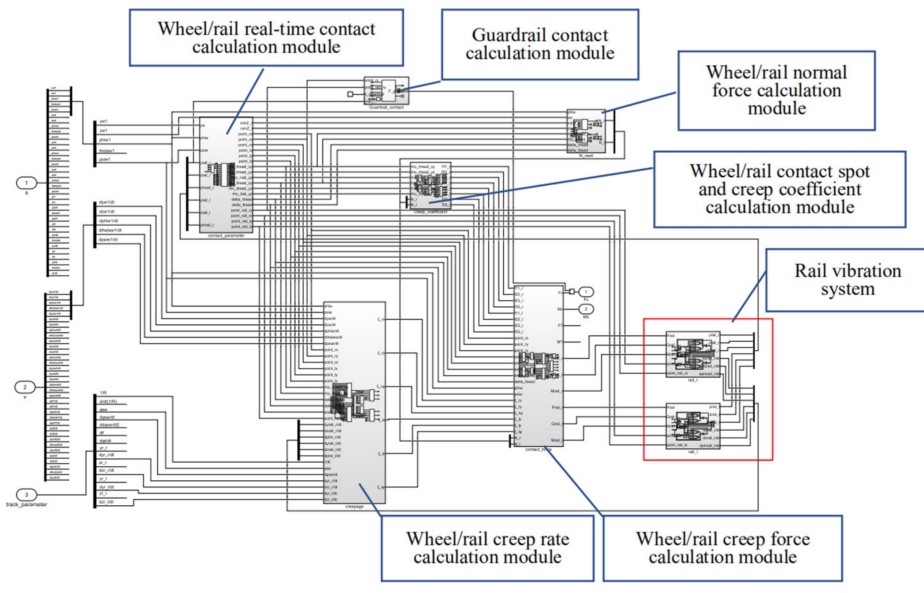

**Figure 6.** Wheel/rail interaction calculation module based on Simulink.

## 4. Geometric Parameter Optimization Design for the Fixed Frog Area

### 4.1. PSO Algorithm

PSO algorithm has faster calculation speed and better global search ability, and each particle will continuously follow its own position and flight speed according to the individ-

ual optimal solution and group optimal solution generated in the iterative process, and then track the present optimal particle to constantly find in the search space according to certain rules, so as to solve the complex optimization problem [22].

According to the particle swarm optimization algorithm, it is assumed that $N$ particles are constantly searching for optimization in a $D$-dimensional search space, and the current position and flight speed of the $i$-th particle are recorded as Equations (6) and (7).

$$\boldsymbol{H}_i = (h_{i1},\ h_{i2},\ \ldots,\ h_{iD}), \qquad i = 1,\ 2,\ \ldots,\ N \tag{6}$$

$$\boldsymbol{V}_i = (v_{i1},\ v_{i2},\ \ldots,\ v_{iD}), \qquad i = 1,\ 2,\ \ldots,\ N \tag{7}$$

The individual optimal solution and group optimal solution of the $i$-th particle are recorded as:

$$\boldsymbol{P}_{\text{best}} = (p_{i1},\ p_{i2},\ \ldots,\ p_{iD}) \tag{8}$$

$$\boldsymbol{G}_{\text{best}} = (g_{i1},\ g_{i2},\ \ldots,\ g_{iD}) \tag{9}$$

Before finding the target solution, all particles update the flight speed and position at the next time according to Equation (10):

$$\begin{cases} v_{ij}(t+1) = \omega v_{ij}(t) + c_1 r_1 \big[ p_{ij}(t) - h_{ij}(t) \big] + c_2 r_2 \big[ g_j(t) - h_{ij}(t) \big] \\ h_{ij}(t+1) = h_{ij}(t) + v_{ij}(t+1) \end{cases} \tag{10}$$

where, $h_{ij}$ is the current position of each particle, $v_{ij}$ is the flight speed of each particle, and $v_{ij} \in [-v_{\max}, v_{\max}]$, $c_1$ is the individual learning factor, $c_2$ is the social learning factor, $p_{ij}$ is the individual optimal solution, $p_j$ is the overall optimal solution, $r_1$ and $r_2$ are the random number between [0, 1], and $\omega$ is the inertia weight.

In order to ensure that the PSO algorithm has a high probability of convergence to the global optimal position, the inertia weight can be adjusted dynamically $\omega$ to balance global search ability and local search ability, as shown in Equation (11):

$$\omega = \omega_{\max} - \frac{(\omega_{\max} - \omega_{\min})}{T_{\max}} t \tag{11}$$

where, $\omega_{\max}$ is the maximum inertia weight, $\omega_{\min}$ is the minimum inertia weight, $T_{\max}$ is the maximum iterations number, and $t$ is the current iterations number.

### 4.2. Optimization Design Method for the Wing Rail Lifting Values

Figure 7 shows the plan view and side view of the wing rail, which deviates to both sides of gauge line with the longitudinal extension of frog. Section A is the throat of the frog, and section K is the section with the top width of the nose rail of 50 mm. Nine key sections are taken between section A and section K, then the wing rail can be interpolated from a total of 11 sections above. Section A is the starting point of deviation and lifting, so the lifting value of the wing rail at section A is set to zero ($h_A = 0$). Based on wheel/rail contact characteristics of the fixed frog area mentioned above, the wheel load transfer has been completed at section K, so the lifting value of the wing rail at section K is also set to zero ($h_k = 0$). The lifting value of the nine sections from section B to section J is set as a series of variable values $h_k$ (k = 1~9), a series of complete wing rail profiles can be obtained by optimizing the wing rail lifting values of the nine sections.

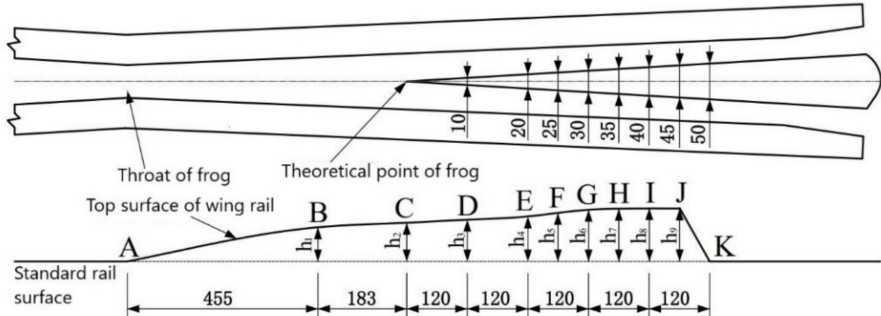

**Figure 7.** Plan and side view of the wing rail.

As the object of optimization are wheel/rail forces at the stock rail and the frog area, the objective function of the optimization model can be determined by Equation (12):

$$\begin{cases} f_1(h_k) = \max(|F_{yli}|) \\ f_2(h_k) = \max(|F_{zli}|) \\ f_3(h_k) = \max(|F_{yri}|) \\ f_4(h_k) = \max(|F_{zri}|) \end{cases} \quad i = 1, 2, 3, 4, \, k = 1, 2\ldots 9 \tag{12}$$

where, $F_{yli}$ and $F_{yri}$ are wheel/rail lateral forces at the stock rail and the frog area, and $F_{zli}$ and $F_{zri}$ are wheel/rail vertical forces at the stock rail and the frog area.

According to the order of magnitude and importance, the above four objective functions are weighted in turn, and the total optimization objective function is shown in Equation (13):

$$f(h_k) = \omega_1 f_1(h_k) + \omega_2 f_2(h_k) + \omega_3 f_3(h_k) + \omega_4 f_4(h_k) \tag{13}$$

where, $\omega_1 \sim \omega_4$ are the weight coefficient corresponding to each objective function.

In order to ensure the smooth transfer of wheelset load from the wing rail to nose rail, the height of the wing rail top surface should be gradually increased until the maxi-mum value is reached. Therefore, the constraint conditions are set as Equation (14).

$$0 < h_k < h_{k+1} < h_{\max} \tag{14}$$

Wheel/rail contact points jump near the section with the width of the nose rail top surface of 20 mm. When wheelset rolls from the wing rail to the nose rail from the throat of the frog, if the wing rail height value at section E is set unreasonably, the nose rail will bear severe wheel/rail impact load at the weak section with narrow top width, so the minimum height value $h_{\min}$ shall be set, as shown in Equation (15).

$$h_4 > h_{\min} \tag{15}$$

The optimized wing rail shall ensure the driving safety of the vehicle passing through the fixed frog area. Therefore, the derailment coefficient and wheel load reduction rate at the stock rail side and the frog side shall meet the following constraints, as shown in Equations (16) and (17):

$$\max\left(\left|\frac{F_{yli}}{F_{zli}}\right|, \left|\frac{F_{yri}}{F_{zri}}\right|\right) < 1 \tag{16}$$

$$\max\left(\left|\frac{\overline{P} - F_{zli}}{\overline{P}}\right|, \left|\frac{\overline{P} - F_{zri}}{\overline{P}}\right|\right) < 0.6 \tag{17}$$

where, $\overline{P}$ is the average wheel load of the left and right wheels of the wheelset.

Therefore, the established optimization model of wing rail lifting values is as shown in Equations (18) and (19).

The objective function:

$$\min : f(h_1, h_2, \ldots, h_9) = f(h_k) \tag{18}$$

The constraint condition:

$$\begin{cases} 0 < h_k < h_{k+1} < h_{max} \\ h_4 > h_{min} \\ \max\left(\left|\frac{F_{yli}}{F_{zli}}\right|, \left|\frac{F_{yri}}{F_{zri}}\right|\right) < 1 \\ \max\left(\left|\frac{\overline{P}-F_{zli}}{\overline{P}}\right|, \left|\frac{\overline{P}-F_{zri}}{\overline{P}}\right|\right) < 0.6 \end{cases} \tag{19}$$

Figure 8 is the flow chart of the wing lifting values optimization algorithm based on PSO. The optimization process is as follows:

(1) Set the basic parameters of particle swarm optimization algorithm and initialize the population, and randomly generate $N$ groups of wing rail lifting values according to the constraints.

(2) A complete set of frog profiles is generated according to each lifting value and the profile of each key section, and the vehicle dynamics are calculated.

(3) If the dynamic calculation results conform to safety conditions, the individual optimal solution and group optimal solution shall be updated according to the quality of the objective function value, and then judge whether the current iteration times have reached the maximum iterations. If the maximum iterations have been met, the optimization results shall be output.

(4) If the dynamic calculation results do not meet the safety requirements or the current iterations number does not reach the maximum iterations, obtain a new group $N$ lifting values according to Formula (14) and (15), and repeat the above calculation steps until the optimal lifting values are obtained.

To verify the effectiveness of the wing rail lifting optimization algorithm, the dynamic performance of a metro vehicle passing through No. 12 fixed frog in the main direction at the speed of 80 km/h is calculated, and the effects of the non-optimized wing rail and the optimized wing rail on the dynamic performance are compared. See Table 2 for the main calculation parameters. According to the technical manual of railway public works design [23], the maximum height value of wing rail $h_{max}$ is set to 5 mm, the minimum height value of key section E is set to $h_{min} = 2$ mm. The basic parameters of the PSO algorithm are set as follows according to the recommended values in literature [24]: population dimension $D = 9$, population size $N = 10$ and maximum flight speed $v_{max} = 1$, individual learning factor $c_1 = 1.5$, social learning factor $c_2 = 1.5$, maximum iterations $T_{max} = 100$, maximum inertia weight $\omega_{max} = 0.8$, and minimum inertia weight $\omega_{min} = 0.4$.

**Table 2.** Principal parameters.

| Parameter | Value |
|---|---|
| Car body mass (kg) | 41,910 |
| Frame mass (kg) | 4060 |
| Wheelset mass (kg) | 1670 |
| Gauge (m) | 1.435 |
| Nominal rolling circle radius (m) | 0.42 |

Figure 9 shows the evolution curve of the objective function value of the wing rail lifting optimization model solved by particle swarm optimization algorithm. It can be seen that when the number of iterations is about 40, the objective function value has tended to be stable. The optimization results for the wing rail lifting values are shown in Table 3.

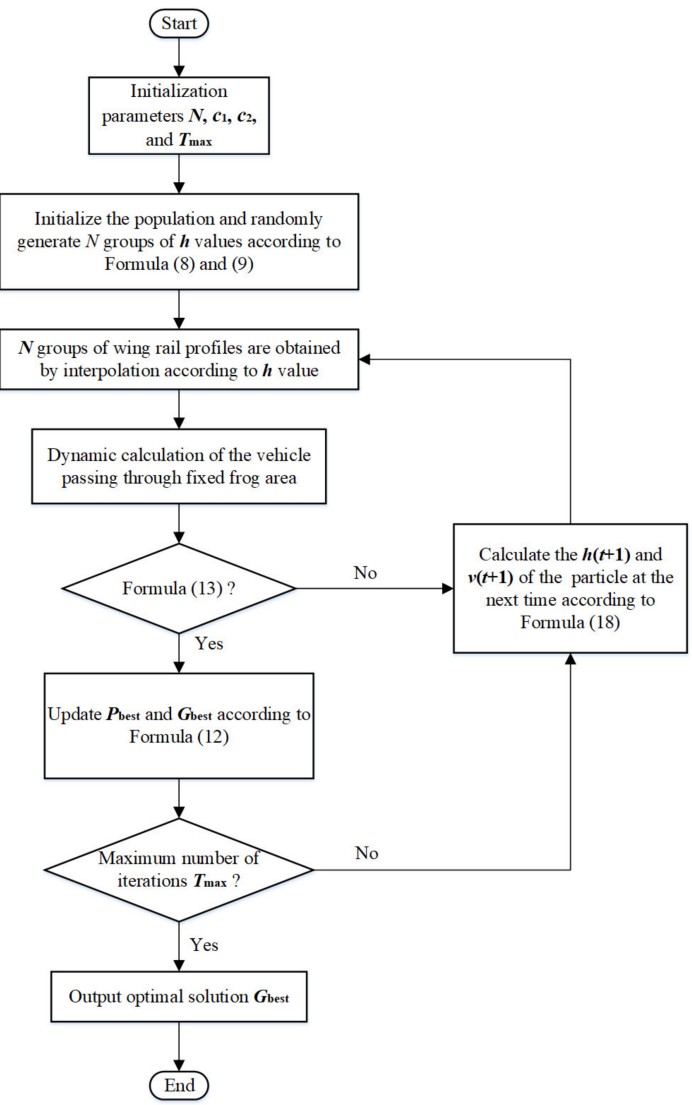

**Figure 8.** Flow chart of the wing rail optimization algorithm based on the PSO.

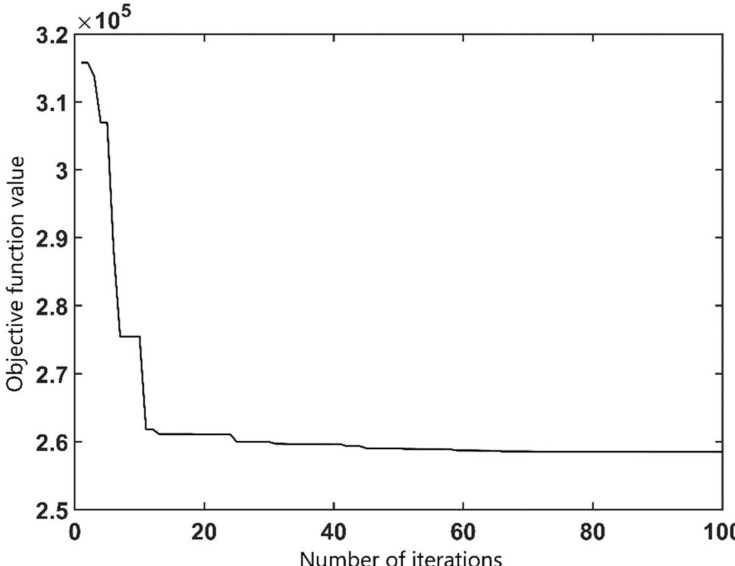

**Figure 9.** Evolution curve of the objective function value.

**Table 3.** Optimization results of the wing rail lifting value of each section.

| Section Position | Optimized Value/mm |
|---|---|
| B | 1.07 |
| C | 2.39 |
| D | 3.81 |
| E | 4.95 |
| F | 4.98 |
| G | 4.99 |
| H | 5 |
| I | 5 |
| J | 5 |

Figure 10 shows the change curve of wheelset vertical displacement of the vehicle passing through the fixed frog before and after optimization. Before wing rail optimization, the maximum vertical displacement of wheelset is 0.78 mm. After the wing rail optimization, the maximum vertical displacement of the wheelset is 0.51 mm, which is reduced by 34.6%, indicating that the optimized wing rail could greatly decrease the vertical vibration amplitude of the wheelset when passing through the frog and improve the running stability of the vehicle.

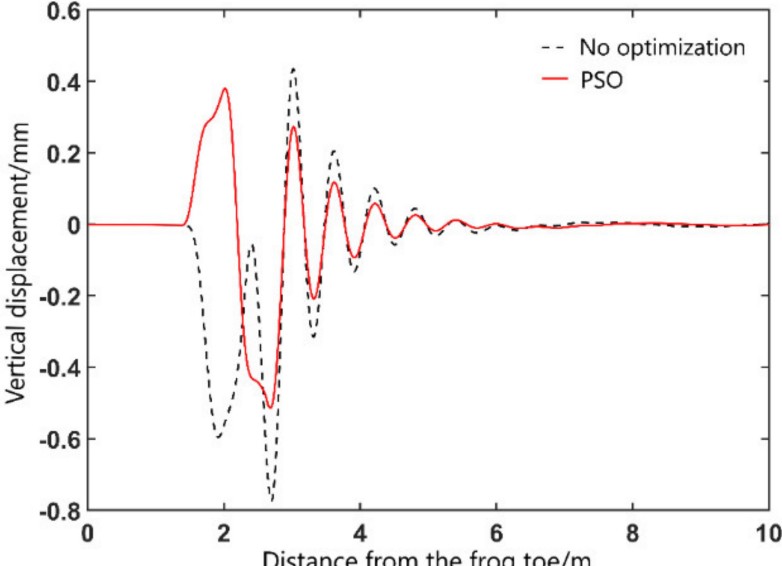

**Figure 10.** Wheelset vertical displacement.

Figures 11 and 12 show the wheel/rail force curve of the vehicle passing through the fixed frog area in main directions. On the stock rail side, after wing rail optimization, the maximum value of wheel/rail lateral force is reduced from 7.8 kN to 6.8 kN, with a decrease range of 12.8%, and the maximum value of wheel/rail vertical force decreases from 83.4 Kn to 75.1 kN, with a decrease of about 10%. On the frog side, after wing rail optimization, the maximum value of wheel/rail lateral force is reduced from 18.4 Kn to 11.4 kN, with a decrease range of 38%, and the maximum value of wheel/rail vertical force is reduced from 104 kN to 91 kN, with a decrease range of 12.5%. To sum up, the optimized lifting value of wing rail can greatly reduce the impact force between the wheel and rail. In addition, based on the wheel/rail force, the maximum value of the wheel load reduction rate is 0.53 and the maximum value of derailment coefficient is 0.2, indicating that the optimized wing rail can ensure the safety of vehicles passing through the turnout in the main direction.

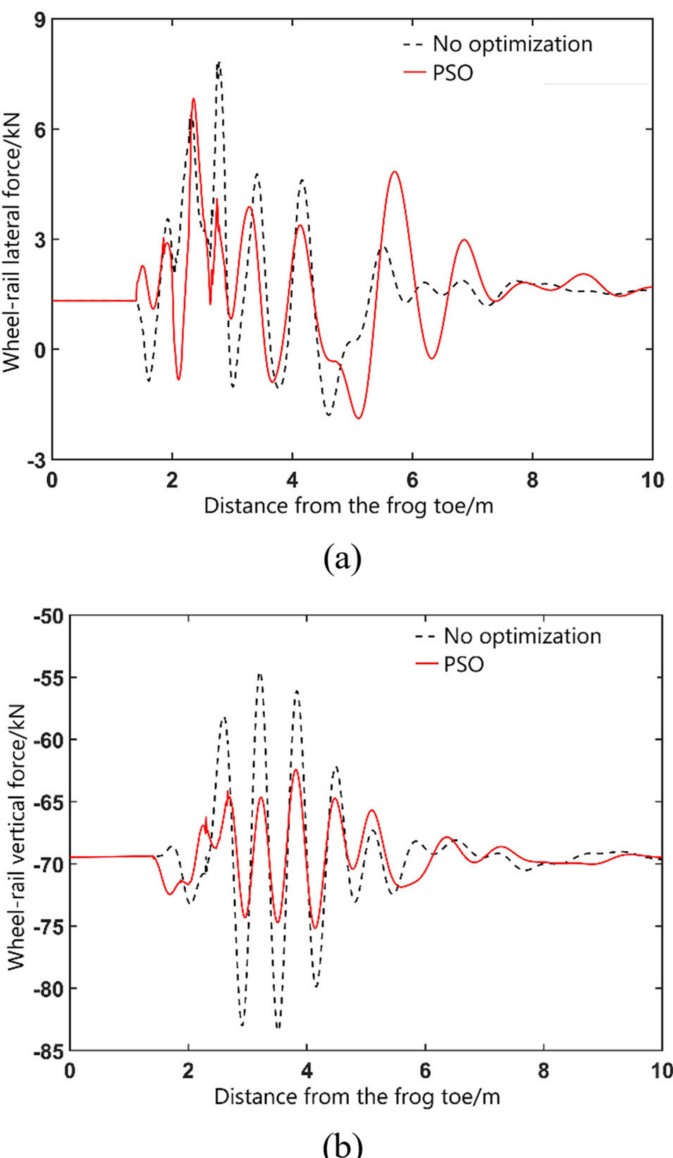

**Figure 11.** On the stock rail side: (**a**) Wheel/rail lateral force; (**b**) Wheel/rail vertical force.

*4.3. Optimization Design Method for the Nose Rail Height*

The plan and side schematic diagram of the nose rail in the fixed frog area is shown in Figure 13. The nose rail is located in the center of the two wing rails, symmetrically distributed relative to the gauge line, and continuously widened and heightened with the longitudinal direction of the frog from the theoretical point of frog, so as to meet the needs of the smooth crossing of wheels.

Section A is the theoretical point of frog, the top width of the nose rail at Section B is 20 mm, and the top width of the nose rail at section H is 50 mm. One section is set every 5 mm from section B to section H, and the profile of the nose rail can be interpolated from the above 8 key sections. Set the height of a total of 6 sections from section B to section G to the variable value $h_k$, $k = 1, 2 \ldots 6$, and set the height values of section A and section H to the fixed value h, respectively, $h_A$ and $h_H$. A series of nose rail profiles can be generated by continuously adjusting the nose rail height values of the 6 sections.

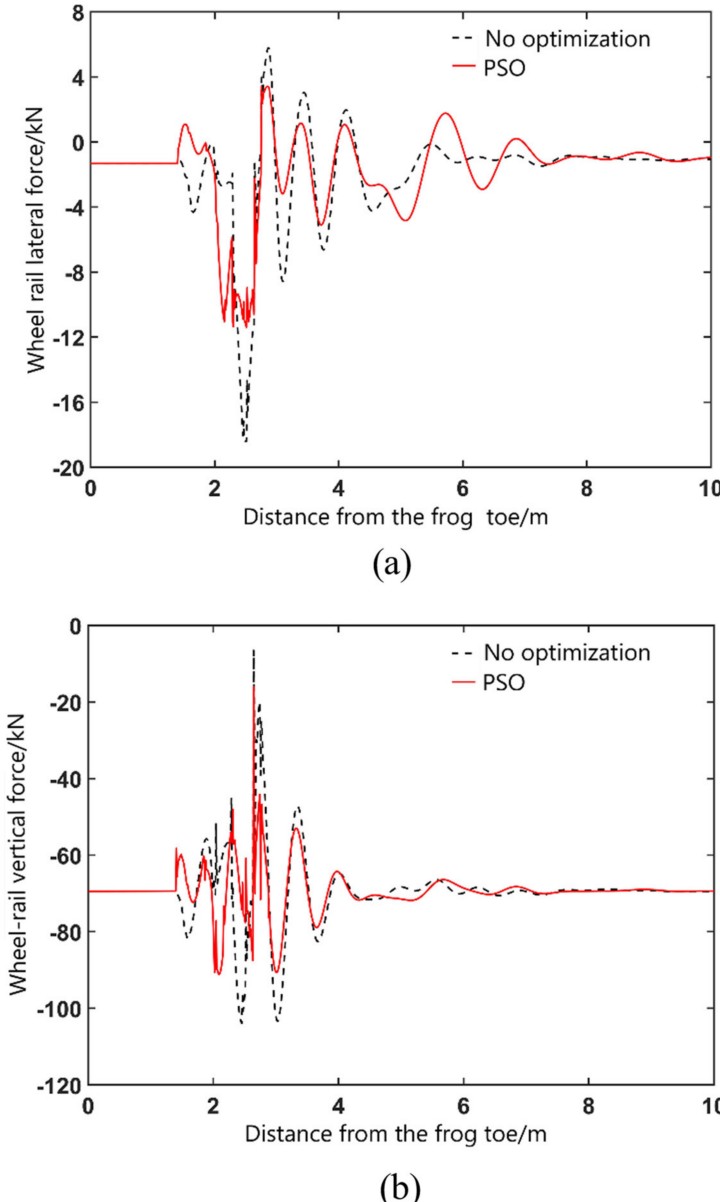

**Figure 12.** On the frog side: (**a**) Wheel/rail lateral force; (**b**) Wheel/rail vertical force.

In the same way, the wheel/rail force at the stock rail and the frog area is selected as the evaluation index of nose rail height. Therefore, the objective function of the optimization model is determined by Equation (12), and the total objective function is determined by Equation (13). To reduce the wheel/rail impact force generated when the wheel load transits from the wing rail to the nose rail, the height of the top surface of the nose rail should be gradually increased until it reaches the standard rail surface height. In order to make the optimized top surface of the nose rail smooth enough, the height value of each key section should be kept decreasing along the longitudinal direction of the frog, with the following constraints, as shown in Equation (20).

$$h_A > h_k > h_{k+1} > h_H \tag{20}$$

Similarly, the minimum height value $h_{\min}$ shall be set according to the bearing capacity of the nose rail, and the height value at section B shall meet the following constraints:

$$h_1 > h_{\min} \tag{21}$$

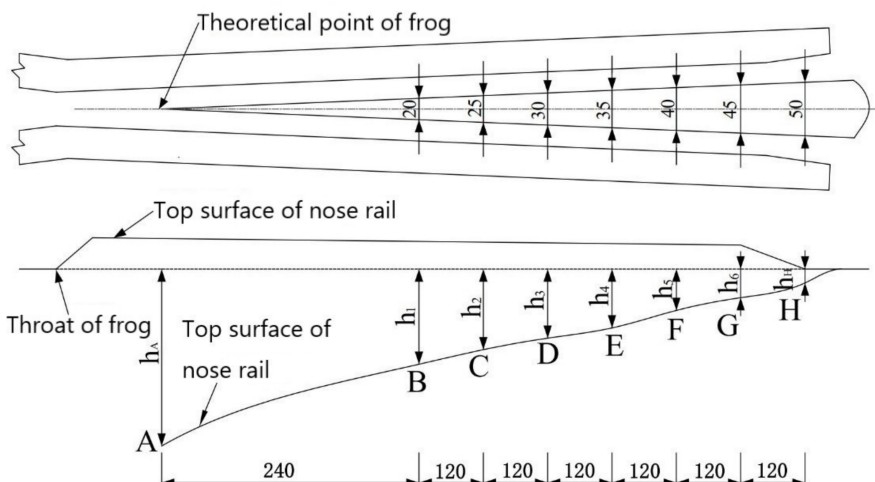

**Figure 13.** Plan and side view of the nose rail.

According to the technical manual of railway public works design, the nose rail height value of each fixed section is set as: the reduction value at Section A is $h_A = 6$ mm, the minimum height value at Section B is $h_{min} = 2$ mm, the height value at section H is $h_H = 0$ [23]. At the same time, the population dimension in particle swarm optimization algorithm is set to $D = 6$, and other parameters are the same as the wing rail lifting optimization design example. The calculation process is similar to the wing rail optimization method and will not be repeated here.

Therefore, the established optimization model of the nose rail height values is as follows.

The objective function:

$$\min : f(h_1, h_2, \ldots, h_6) = f(h_k) \tag{22}$$

The constraint condition:

$$
\begin{cases}
h_A > h_k > h_{k+1} > h_H \\
h_1 > h_{min} \\
\max \left( \left| \frac{F_{yli}}{F_{zli}} \right|, \left| \frac{F_{yri}}{F_{zri}} \right| \right) < 1 \\
\max \left( \left| \frac{\overline{P} - F_{zli}}{\overline{P}} \right|, \left| \frac{\overline{P} - F_{zri}}{\overline{P}} \right| \right) < 0.6
\end{cases} \tag{23}
$$

Figure 14 shows the evolution curve of the objective function value of the nose rail height optimization model solved by particle swarm optimization algorithm. It can be seen that when the number of iterations is about 85, the objective function value has tended to be stable. The optimization results of the wing rail heightening value are shown in Table 4.

**Table 4.** Optimization results of the nose rail height value of each section.

| Section Position | Optimized Value/mm |
|:---:|:---:|
| B | 3.22 |
| C | 1.98 |
| D | 1.65 |
| E | 1.49 |
| F | 1.23 |
| G | 0.72 |

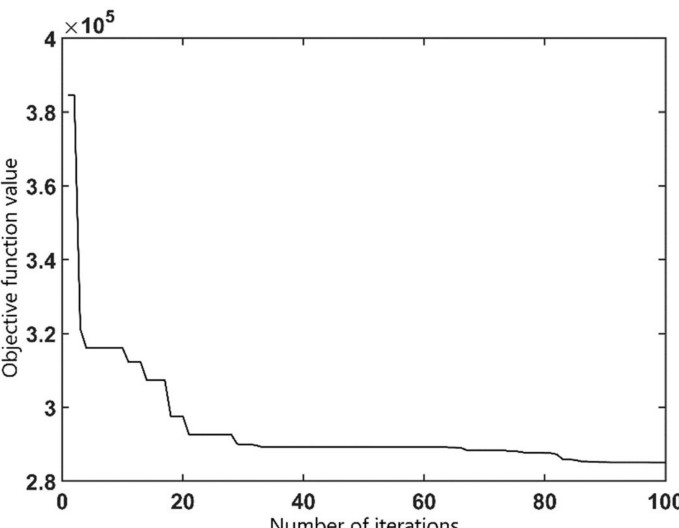

**Figure 14.** Evolution curve of the objective function value.

Figure 15 shows the change curve of wheel vertical displacement of the vehicle passing through the fixed frog before and after optimization. Before the nose rail optimization, the maximum vertical displacement of wheelset is 0.77 mm. After optimization, the maximum vertical displacement of the wheelset is 0.62 mm, which is reduced by 19.5%.

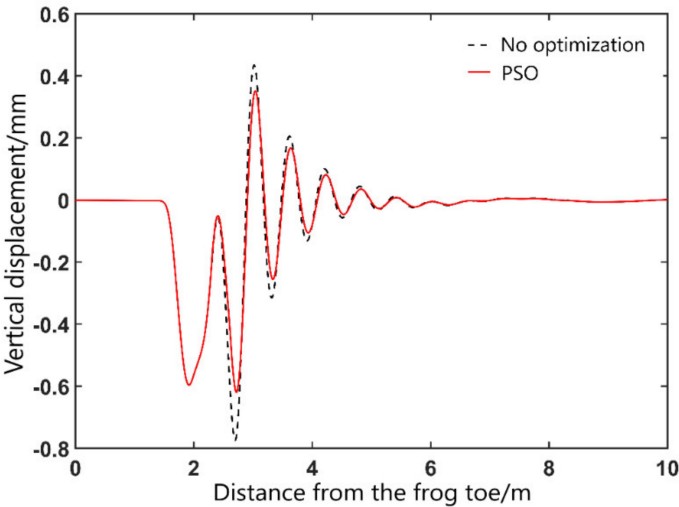

**Figure 15.** Wheelset vertical displacement.

Figures 16 and 17 show the wheel/rail force curve of the vehicle passing through the fixed frog area in main directions. On side of the stock rail side, after the nose rail optimization, the maximum value of wheel/rail lateral force is reduced from 7.8 kN to 6.6 kN, with a decrease range of 15.4%, and the maximum value of wheel/rail vertical force decreases from 83.4 kN to 80.5 kN, with a decrease of about 3.5%. On the side of the frog, after wing rail optimization, the maximum value of wheel/rail lateral force is reduced from 18.4 kN to 15 kN, with a decrease range of 18.5%, and the maximum value of wheel/rail vertical force is reduced from 106 kN to 96 kN, with a decrease range of 9.4%. To sum up, the optimized height values of nose rail can also reduce the impact force between the wheel and rail. In addition, according to the change curve of wheel/rail force, it can be calculated that the derailment coefficient and wheel load reduction rate are within the safe range, that is, the optimization algorithm does not affect the safety of the vehicle.

The effectiveness of the optimization method is verified by optimizing the wing rail lifting value and nose rail height of the fixed frog area. By analyzing the wheel/rail

dynamic response in a fixed frog area before and after optimization, it is found that the optimized fixed frog can significantly improve the wheel/rail relationship in the fixed frog area, reduce the vertical impact force and lateral force of the wheel/rail, improve the running stability and driving safety of the vehicle, and thus extend the service life of the fixed frog. At the same time, by comparing and analyzing the optimization results of the wing rail lifting value and nose rail height, it is found that the optimization of wing rail lifting value improves the wheel/rail interaction performance in the frog area more.

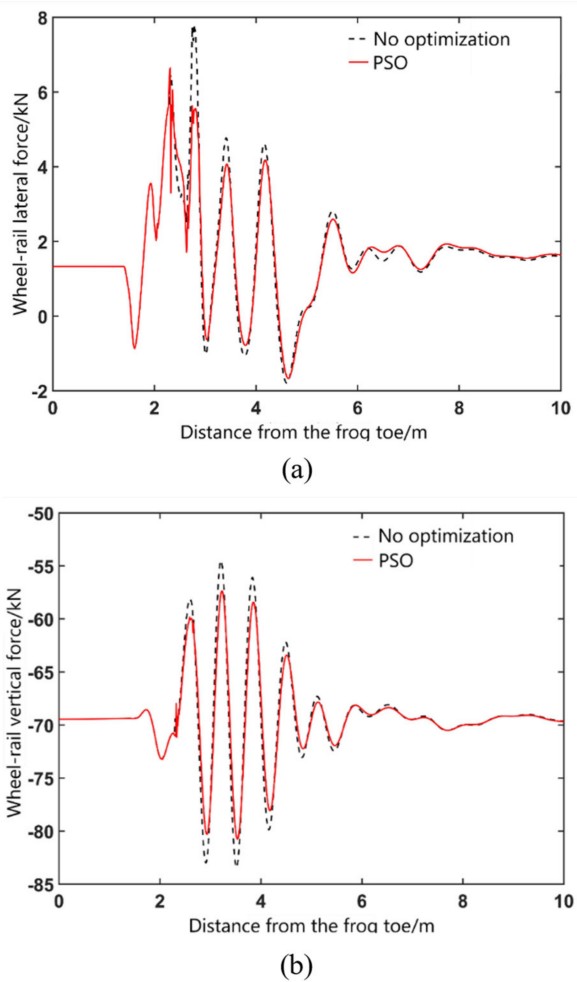

**Figure 16.** On the stock rail side: (**a**) Wheel/rail lateral force; (**b**) Wheel/rail vertical force.

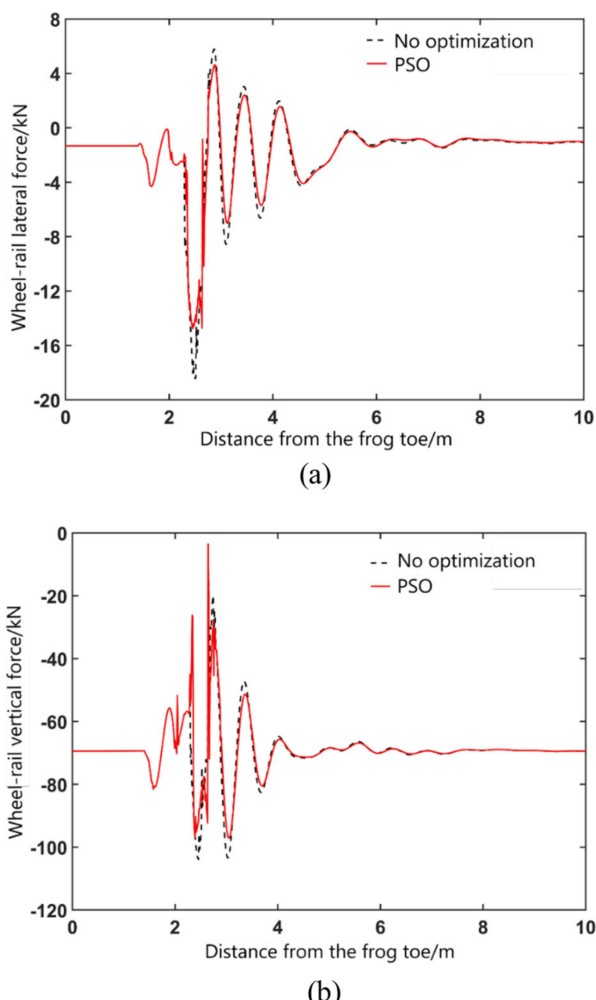

**Figure 17.** On the frog side: (**a**) Wheel/rail lateral force; (**b**) Wheel/rail vertical force.

## 5. Conclusions

In this paper, the wheel/rail contact characteristics in the fixed frog area are comprehensively analyzed, the optimization models of wing rail lifting values and nose rail height values in the fixed frog area are established, and the specific optimization design process based on particle swarm optimization algorithm is proposed. The main conclusions are as follows:

(1)  If the wheelset is close to the nose rail with a large lateral displacement, there is a risk that the wheel will climb onto the nose rail. In order to reduce the vertical impact of wheelsets passing through the fixed frog area and improve the safety and stability of vehicles passing through the turnout center, the geometric parameters of the fixed frog should be set reasonably.

(2)  After wing rail lifting values optimization, the maximum vertical displacement of the wheelset is reduced by 34.6%. On the stock rail side, the maximum wheel/rail lateral force is reduced by 12.8%, and the maximum wheel/rail vertical force is reduced by 10%. On the frog side, the maximum wheel/rail lateral force is reduced by 38%, and the maximum wheel/rail lateral force is reduced by 12.5%.

(3)  After nose rail height values optimization, the maximum vertical displacement of the wheelset is reduced by 19.5%. On the stock rail side, the maximum wheel/rail lateral force is reduced by 15.4%, and the maximum wheel/rail vertical force is reduced by 3.5%. On the frog side, the maximum wheel/rail lateral force is reduced by 18.5%, and the maximum wheel/rail lateral force is reduced by 9.4%.

## 6. Future Work

The optimization algorithm proposed in this paper by using original PSO has a good effect on the optimization of geometric parameters of the fixed frog. However, with the continuous improvement and development of the random search algorithm based on group cooperation, there will be more options to solve the optimization problem of the fixed frog geometric parameters in the future, such as the Grey Wolf Optimizer (GWO), Sine Cosine Algorithm (SCA), Harris-Hawk Optimizer (HHO), Whale Optimization Algorithm (WOA), Arithmetic Optimization Algorithm (AOA), and their hybrid algorithm. In subsequent research, on the basis of the above algorithms, by comparing the characteristics of different algorithms, we can explore better optimization algorithms or hybrid algorithms to solve engineering problems.

**Author Contributions:** Methodology, validation, writing—original draft preparation, R.Z.; supervision, writing—review and editing, G.S.; investigation, software, X.W. All authors have read and agreed to the published version of the manuscript.

**Funding:** This research was funded by the Science and Technology Research and Development Plan of China Railway Corporation (Grant No. 2017G003-A).

**Institutional Review Board Statement:** Not applicable.

**Informed Consent Statement:** Not applicable.

**Data Availability Statement:** The authors confirm that the data supporting the findings of this study are available within the article.

**Conflicts of Interest:** The authors declare no conflict of interest.

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
