# Peer review of "Research on Geometric Parameters Optimization of Fixed Frog Based on Particle Swarm Optimization Algorithm"

_applsci, doi:10.3390/app122211549_

Round 1
Reviewer 1 Report
The article is very interesting. The subject is important because the research results may contribute to increasing the safety during the operation of rail vehicles.
My comments to the article
1. Figures 4 and 8 are illegible, it is not possible to assess the correctness of the model on their basis.
2. In Figure 8, the variables have no equivalents in the systems of equations (1) and (3).
3. The force Fpr is not marked in Figure 5
4. There are errors in the system of equations (1). If they were transferred to the model, the results do not make sense, if they were only made in the record, then a correction should be made.
5. In the system of equations (3), the letter M has two meanings, one means mass and in another equation, a moment of force.
6. The algorithm in figure 10 is unreadable, additionally the End module is missing.
7. I believe that the forces occurring during crossing points have a large impact on the wear of wheels and rails, so it is worth adding to the bibliography
https://doi.org/10.1016/j.triboint.2020.106365
Reviewer 2 Report
This paper presents a metaheuristic application to improve the wheel-rail interaction performance of the rail vehicle passing through the fixed frog area based on a geometric parameter optimization design method of the fixed frog area and particle swarm optimization. The paper is interesting and relevant. However, the paper needs some improvements before acceptance as follows:
1- The structure of the article is not organized properly. The literature review is mixed with the theoretical background and the introduction is very brief. It is recommended to rewrite the introduction section and separate the literature review from the theoretical background.
2- Some figures might be copied from other sources. Please obtain the necessary copyrights and cite accordingly.
3- The railway vibration system in this paper is a simplified model. What are the other alternatives? Please discuss!
4- Results discussion is relatively not mature. Please further improve your discussion or add it as a separate section.
5-The authors have used the original PSO for their optimization problem. But they have missed some recent advances in PSO improvements and hybridizations. The authors should discuss the use of more advanced algorithms or consider comparing them as future work such as the combined social engineering particle swarm optimization as an improvement of PSO, or other new metaheuristics such as the new self-adaptive quasi-oppositional stochastic fractal search.
6- Future work can be added to the conclusion section. Some future improvements such as the comparison with other original metaheuristics such as the Grey Wolf Optimizer, Artificial Bee Colony, Sine-cosine algorithm, Harris-Hawk Optimizer, Whale optimization algorithm, Arithmetic optimization algorithm, etc., and other hybrid algorithms such as the aforementioned algorithms and other representative hybrid algorithms in the literature. Moreover, the authors can improve their work based of proposing their own hybrid methods.
